# Framing Sustainable Healthcare Services

**DOI:** 10.3390/ijerph18126336

**Published:** 2021-06-11

**Authors:** Per Engelseth, Remiguisz Kozlowski, Karolina Kamecka, Lukasz Gawinski, Richard Glavee-Geo

**Affiliations:** 1Tromsø School of Business and Economics, University of Tromsø, Narvik Campus, 8505 Narvik, Norway; 2Center of Security Technologies in Logistics, Faculty of Management, University of Lodz, 90-237 Lodz, Poland; remigiusz.kozlowski@uni.lodz.pl; 3Department of Management and Logistics in Health Care, Faculty of Health Sciences, Medical University of Lodz, 90-131 Lodz, Poland; karolina.kamecka1@stud.umed.lodz.pl (K.K.); lgaw@agumed.edu.pl (L.G.); 4Department of International Business, Norwegian University of Technology and Science, 6009 Ålesund, Norway; rigl@ntnu.no

**Keywords:** healthcare, sustainability, logistical operations, ecosystems, framing, overflowing, exchange economy, integration, interdependency, process thinking

## Abstract

This paper develops an analytical framework using process thinking to achieve sustainable healthcare services. Healthcare is characterised by low economic efficiency. At the same time, it is embedded in ethical concerns related to society and nature. Healthcare is thus conceptualised as functionality in an ecosystem. The patient is woven into both nature and society. Given the complex nature of healthcare services, we seek an alternative way to understand healthcare services, focusing on the exchange aspect of the economy. We offer a conceptual model that helps build an analytical framework focusing on how practitioners and leaders in healthcare frame their activities. This framing provides guidance in healthcare practice. Furthermore, framing is associated with both healthcare service providers as well as patients and next of kin—the recipients. This framework aims to guide practical research and development activities in healthcare.

## 1. Background and Introduction

Today’s healthcare management binds large amounts of time and resources. This may be characterised as wasteful, calling for approaches dedicated to solving the economics of providing quality healthcare. Healthcare management normally follows a predominantly deterministic approach, implying that budgets created from past registered experiences are used to plan healthcare processes through a combination of intuition and often archaic and difficult to use computer software [1,2]. Healthcare services often have a high level of criticality and they are also intertwined with ethical issues of human well-being. This underpins the importance of developing healthcare services for a societal perspective. This also points to that these services need to be sustainable. Healthcare services emerge as continuously cognition through trial and error processes. Healthcare feature interactions between the patient, their next of kin and the many different types of healthcare professionals involved in the diagnosis and treatment. This influences the predictability of these services. Healthcare as a process, is therefore always, to some degree, emergent [1,2]. Optimisation is therefore hard to plan in healthcare. Process improvement and a search for better quality is, however, very important in achieving sustainable health care services. A notion of continuous improvement is therefore vital.

This paper aims to develop an analytical framework that applies process thinking to achieve sustainable healthcare services. This approach includes applying ecosystems thinking to developing practices in healthcare management. Ecosystems are a variation of systems thinking. It highlights the environmentally contingent and the therefore also the complex nature of achieving sustainable healthcare services. The developed approach guides the researcher in seeing healthcare as an emergent process. This makes this approach not subject to development tools applying deterministic planning procedures. White states, regarding this approach, that “health, as a feature of human well-being, is actually an expression of people embodied in ‘nature’, although nature itself certainly does not automatically bestow good health” [3].

Likewise, this approach directs attention to managerial perception of this healthcare service development. Hence, directing attention to how managers frame the complexity of attaining sustainable healthcare services provided in the broader context of societal and environmental needs. Subsequently, we define framing as a cognitive activity where people make sense of the ever-changing reality surrounding them by constructing simple activity-directing mental models. Furthermore, including assessment of contingencies, healthcare’s environmental needs are not limited to economic concerns, but also that healthcare concerns both societal ethics and nature-related biology concerns. By pointing to this process-founded people-in-context approach to research healthcare, this conceptual development further develops preceding conceptual research [1,2] by creating a stronger focus on how to do such research in healthcare in scientific practice. This conceptual development includes combining this approach with conceptual understandings on networking in complex production processes [4,5,6]. The focused research problem concerning how to study sustainable healthcare services at hand is accordingly: Providing a process-based analytical framework focusing on how networked actors dynamically frame sustainable healthcare service provision. This endeavour seeks to guide those carrying out research as well as service development. The application of “sustainability” underpins the importance of achieving continuous improvement in healthcare service provision. It is in this mainly the societal aspect of sustainability that is in force notwithstanding that people are biological beings embedded in both nature and an economy.

This approach’s choice is grounded on the reasoning that healthcare services face a diverse range of quality challenges. In this paper, such services are regarded as economic production activities. This reasoning does not weaken the concern that healthcare is a prime feature of human well-being, and therefore clearly ethically laden. To create quality healthcare, it is vital that healthcare services are produced in a manner that economically uses societal resources and at the same time creates a process output that sustains the welfare of the treated patients. Kritchanchai et al. [1] point out that healthcare service is an economic activity that can be improved through applying logistics and supply chain management approaches. Engelseth and Kritchanchai [2] created as shown in Figure 1a strategic model of healthcare service development that depicts the focal nature of processes when studying healthcare services and environmentally contingent production activities:

This figure is fundamental to the line of thinking that guides this research. It shows how, from an ecosystem’s viewpoint, healthcare is an environmentally contingent form of economic production. Engelseth et al. [4] point out how healthcare services to a varying degree are complex. Based on three empirical cases of healthcare services, the model conceptually suggests that this form of production may be improved through simulation based on this modelling. What is decisive is that healthcare planners and developers should not only aim to manage healthcare services following a deterministic planning paradigm but rather, in line with Engelseth and Kritchanchai [2] and Engelseth et al. [5] as a fundamentally complex system. Here we aim to provide practical guidance into the grounds for as well as how to accomplish this research feat.

Healthcare in this process-based approach is fundamentally viewed as complex operations [7,8]. Processes are dynamic in nature, concerned with pooling and transforming resources. In services, the output is not a physical product but rather a more abstract fulfilment of customer expectations. To produce, these logistical operations in healthcare are a series of interconnected tasks that need to be coordinated to achieve efficiency. There is both a horizontal (between similar or possibly competing actors) and a vertical (between dissimilar complementary actors) aspect to this coordination. Processes, since embedded in networks, are thus inherently complex and therefore emergent [7]. The time used as well as the timing (coordination) of the process, involve analysing production as a mixture of objective and subjective meanings [9]. This means that we aim to follow healthcare as a sustainable production system consisting of numerous interacting ‘pieces’ in processes embedded in a common healthcare network of interacting organisations. By process thinking, we consider the ‘healthcare network’ as a dynamic phenomenon–in terms of movement, activity, events, change and temporal evolution. The healthcare network involves considering how and why things (e.g., people, organisations, strategies, environments) change, act, and evolve temporally in a complex and dynamic progression [10].

In taking an ecosystems approach, this also means that healthcare is embedded in features of both nature and society as essential aspects of this economical value-producing environment. Finally, the importance of sustainable healthcare is grounded on the fundamental assumption that if such service production is effective and efficient in a long-term perspective, the analysis must heed that healthcare services are importantly environmentally contingent. In the following sections, we discuss the main elements of this approach, starting with the concept of sustainability. This is followed by considerations regarding ecosystems, network structure, process and framing healthcare services. In conclusion, a conceptual model to guide research on understanding and developing environmentally contingent framing of healthcare services is provided.

## 2. The Foundation: “Sustainability” Discussed

The concept of sustainability was first articulated in the Brundtland Report of 1987 with the objective of conserving resources for the use of future generations. Sustainability can be viewed as either strong or weak [11]. While a strong view encompasses mandatory demands to pass on the same stock of environmental assets from the current to the next generation, the weak view opens for the second generation to inherit some substituted environmental assets by human-made ones from the previous generation as long as the total wealth is conserved [12]. This assessment of stainability is not entirely objective. For instance, while air quality may call for a strong viewpoint, other issues may call for the weak one, e.g., fossil fuel consumption. Strong and weak sustainability can thus be understood as hard and soft constraints, respectively.

According to Engelseth et al. [13]: “‘Sustainable production’ archetypes have progressed towards a triadic structure of solutions, grounded in the three dimensions of sustainability: economy, ecology and equity”. It is important to discern between “sustainability” out there, in reality, so to say, and how people conceive “sustainability” conceptually. The latter is an expression of societal discourse and is a mental model subject to change through societal interaction. This reasoning implies there should be scrutiny directed not only to how societal actors achieve a greener society but also how they discuss and perceive this achieving by scrutinising the applied mental models of “sustainable production”.

Conceptual models on sustainable production [14] represent the interrelationship between the environmental, economic and social aspects of sustainability. This interdependence between nature, society and the environment is, however, complex [15,16,17]. Singh et al. [16] point out that people in business and society develop and use conceptual models to simplify reality. These models are tools helping to generate more easily understandable mental models, thereby supporting discourse, the generation of ideas on how to attain sustainable production. These models provide managers with a systems perspective of sustainable production that encompasses three aspects: nature, society and economy [18]. The most widespread model on “sustainability” is relatively static. It is derived from the “Triple Bottom Line” (TBL) model and conceptualised by Elkington [19]. TBL provides a basic analytical framework to manage sustainability in large and complex manufacturing projects [20]. The model described by Lozano [21] is shown in Figure 2 below:

This model of sustainability amplifies several challenges like the rapid changes in markets, legislation, technologies, and customer demands [22]. Kuhlman and Farrington [23] show how this model also clarifies that both the social and economic dimensions should be combined in a single socio-economic dimension termed “well-being” to avoid obscuring the role of the desire for a better life and how this impacts the environment. There are incentives that help implement and improve these two or three dimensions. While economics is governed by the market and driven by the profitability incentive, environmental and social dimensions are regulated by standards, e.g., ISO 14001 and Social Accountability SA 8000. Other standards and initiatives are field specific and are used as preliminary steps towards meeting the requirements of the more general, prestigious, and international certifications. The latter constitute a competitive advantage and generate goodwill businesswise.

Based on a literature review of 191 scientific papers on sustainable supply chain management, Seuring and Müller [24] point out that while the focus is mainly on environmental issues, the social dimension and integrative sustainability are beginning to emerge in this inter-organisational approach to doing business. Simpson and Power [25] provide, based on a literature review, a conceptual framework integrating concepts of (1) supply relationship, (2) lean manufacturing, and (3) environmental management practices.

At a more general level, Lozano [21] points to an alternative model of sustainable production consisting of three concentric spheres; the ‘economic’ and ‘social’ spheres conceptually understood as subsystems shown in Figure 3:

These graphical spheres depict the economic sphere as reliant on the ‘healthiness’ of the first, societal, and environmental sphere. While the interlocking circles model of sustainability depicts the suitability as the intersection of economy, society and environment, the concentric circles model presumes that a healthy economy is dependent on features of society and nature. This underpins that there are alternative ways to view sustainable production. The triple bottom line does evoke the awareness of sustainability as an important aspect of production. It remains on a level where sustainability is a value negotiated. The concentric circles model goes a step further since it illustrates that production is embedded in society and nature. Features of society and nature continuously affect production regardless of negotiation.

The concentric circles model depicts production as economic activity, a process embedded in society and culture. When taking a process view, achieving sustainable production depends on operations in context and the wider environment. It is, therefore, necessarily an emergent process in a complex production system. When directing the view to the production of healthcare services, developing this form of production must consider features of organising various healthcare processes themselves and how these processes are embedded in society and the wider environment. It also heightens the importance of ethics, the long-term human well-being. The next step is to consider this fundamental view in light of an ecosystem approach.

## 3. Ecosystems: Functional Interconnectivity

Haeckel conceptualised ecology in 1866 as the science of relations between the organism and the surrounding outer world [26]. This highlights the contingent nature of all the outgoings in the world, including production activities. In this picture, “ecosystems” involves highlighting theoretically the interdependence between nature, society and business applying a system’s perspective. Systems thinking is the cornerstone of an ecosystems approach.

Systems are here understood as networks of resources displaying economic activity when they are pooled. The system always has an overarching function, and it is encompassed by a more or less, to the actors involved, clear borderline; thus, and concerns the degree to which a system is open or closed. The system has a clear overarching function and does not eliminate the feature of nested sub-functions. These sub-functions are derived from the main function of the system. The systems approach is a distinct way to simplify reality, conceptually model it, and thereby analyse it. The value of the systems approach lies in its resource pooling activity, the operations, create synergies. The outcome of pooling resource is not equal to the value of keeping systems in isolation without integration. It should be more, but this is not guaranteed since any logistical operation may “go wrong”.

Compared with a network approach, where interconnectedness is a core feature of resources, activities and actors, the systems approach hampers creativity by not scientifically challenging given functionality, resources, actor features, and finally, the system borders. While a systems approach considers small systemic loops of optimisation within the given limits of analysis, it may be limited depending on the degree of open system approach and innovation quests.

The ecosystems approach is a variant of open systems thinking. It is open because it operates within its systemic borderlines and is influenced by the unruly features of nature and society. It finds its roots in the natural sciences, based on observations of how biological organisms function. Capra and Luisi [27] state that “nature does not show us any isolated building blocks, but rather appears as a complex web of relationships between the various parts of a unified whole”. Systems are found in nature regardless of the eyeglasses the researcher chooses to wear. In this picture, production functionality is embedded systemically in nature.

An ecosystem understanding of healthcare services as a process implies using systems thinking encompassing economic, societal and nature concerns related to producing this service; an expansion of systems border that entails increased complexity. It also expands service production functionality from a narrow economic view to encompass the wider conceptual understanding of sustainability in producing healthcare services. Therefore, healthcare is not merely viewed as a system. It is also interwoven with people in society as well as features of nature.

## 4. Network Structure: Interdependencies and Integration

The network is described in Figure 1 as the intermediary layer between the process (production) and the environment (nature and society). The network structure of healthcare services describes the more static features of this form of production. Network contingencies are described as sets of business relationships. In these relationships, actors commonly do have a face dependent on the degree of integration. Interdependency involves considering the reasoning for this collaboration. Why do networked actors collaborate to produce systemically? Following Thompson [28], interdependencies may also vary on several dimension. Firstly, the nature of interdependency is at a superficial level dependent on the type of industry. Patterns of interdependence vary between industries. According to Thompson [28], services are mainly reciprocally or pooled interdependent, while manufacturing is more sequentially interdependent. In services, the main feature of the development, and this importantly concerns healthcare provision, lies in increasing standardisation of resources to better pool resources or develop negotiations to improve mutual adjustments typical of reciprocal interdependency. The following (see Figure 4) illustrates roles of interdependency in relation to coordination:

In some cases, healthcare services may have strong sequential interdependency, and then development should concern mainly improving coordination over a production timeline. Importantly, all forms of production exhibit all forms of interdependency. The strength of these interdependencies in relation to each other may, however, vary. Economising production is found thus in two realms: process and network context. Economisation of healthcare service engineering services involves changing (1) the context of these healthcare service processes, (2) the supporting context of these operations, and (3) how the process itself is organised involving resource use in operations to produce.

Viewing production as an output of pooled resources managed by networked actors can easily be conceptually fitted into an ecosystem understanding of production. This describes the core process, the operations that produce, e.g., healthcare services in an environmentally contingent manner. This section’s focus is that this immediate context of production is relatively stable and characterised by interdependency and integration. The network structure is a pattern; it is only slowly changing. As Ford and Mouzas [30] state, these patterns of interaction are the result of choices made by actors in the network. Baraldi et al. [31] point out features of networked resources, pointing to how they are complex. Therefore, the network structure is also dynamic but at a different rate of velocity than the processes mainly characterised by change. The network may at any given point of time be measured due to this slow pace of change. Likewise, it is meaningless to measure a process at any given time. It must be measured following a timeline, within a time frame.

Features of pooling resources to produce healthcare services also need to be considered a structural aspect of networked-based production. Håkansson and Snehota [32] point out that resources have a provision and likewise a use side. Complexity is in part associated with the friction between the provision and uses aspect. Marketing is all about bridging this potential gap in perceiving the value in a potential saleable market offering. Value is negotiated in exchanges. The outcome of this negotiation in a network is hardly predictable. Also, pooling resources may be done in many ways. Likewise, different sets of maybe even competing actors may combine and use resources. This actor-perceived aspect of value is emergent in this negotiated networking reality of value creation [33]. This is associated with the receiving actor who values the provided and used resource and is, similarly to the supplier, embedded in a context. Customer value is clearly a moving target for marketers. Interdependency is accordingly a perception by actors of reasoning why they may want to interact with other actors in a network. For a supplier, customer value is always a moving target. Maybe it is easier not to consider “value” as a target but as a critical dialectical element of economic discourse.

Fundamentally, actors involved in producing thought the pooling and using resources in networks are to the degree they are heterogeneous entities (the resources and the actors) harnessing the economies of complementarities. Through integration, mainly enveloped in the use of business relationships over time, production is economised. Likewise, the business relationship is from a network understanding of production considered to be a resource in itself, conceptually separate from the bilateral actors involved [31,32]. Network heterogeneity is a core assumption providing the reason why interaction takes place. This heterogeneity which is a feature of interdependence and integration describes the context’s quality to produce. Networked companies are interdependent due to this heterogeneity, and following Emerson [34], this also explicates network power for how things get done, rather than a foundation for coercive behaviour. Therefore, Ford et al. [35] state that: “The manager becomes someone who must operate within multiple dependencies”. This understanding of network structure provides the foundation for considering how interaction changes at a slower pace and impacts and is impacted by production processes.

## 5. Process: Interactions to Produce

This study focuses on healthcare as a process. This is the unit of analysis. It is the phenomenon subject to development. Processes are embedded in a network context, or a systemic “supply chain” consisting of actors bound together by relationships. Pettigrew [36] defines process as: “A sequence of individual and collective events, actions and activities unfolding over time in a context.” Processes unfold through synchronised, sequentially interdependent decision-making events. As illustrated in Figure 1, the network is the context of the process. A network is interconnected by mutual interactions unfolding continuously in specific and changing contexts [36,37,38]. On these grounds, Cantu et al. [39] state that “solutions (i.e., resource combinations) always emerge as a consequence of interaction among actors. This interaction makes the actual combination dependent on the web of actors involved and, therefore, difficult to predict”. The interaction may be therefore considered as an emergent process. This view is rooted in that (1) solutions (the outcome of production) are continuously interpreted differently by different actors in the network, (2) also actors often have dual roles, as a resource provider and also user, (3) these dual perspectives of actors as resource user and provider are challenged in the context of the business relationship through interaction. Production as value creation is thus also a form of continuous learning. We postulate this as an aspect of the “production” concept.

There are two important aspects of value creation in any form of business network. Following Hammervoll [40], these are (1) production and (2) exchange. Exchanges are managerial activities that bond through relationships in a network. The exchange also takes place between departments in larger firms. Mainly, however, the term exchange point to inter-organisational behaviour. Exchange (management and interaction between managers) supports the production of value (operations). Exchange is a service process in the marketplace. It is associated with achieving value. Through operations’ various processes, production can be considered an economic measure of resource use efficiency and process outcome. Hammervoll [40] means that exchange is likewise as production, an economic entity. The relationship between exchange and production features the managerial exchange support producing.

On these grounds, Janusz et al. [5] develop an analytical framework where process development involves knowing: (1) the business relationship interdependencies (2) how well integrated the network is. Together, features of interdependency and integration, the supply network context, impact on interaction exhibiting for analysis how value is produced and thereby create valued service process outcome. Founded on the classification of interdependency, integration, and interaction as key features of services supply provided by Janusz et al. [5], this understanding of healthcare as an emergent process in a service-producing network is conceptualised as shown in Figure 5:

A process-focused view is a higher-order conceptual view implying living with complexity. This is an alternative to management following deterministic planning and control thinking. A process-focused view enhances perceived complexity where interaction is highlighted as continuous input to decision-making. Processes themselves generate uncertainty through interaction. This, following Kreye [41], reducing uncertainty to economise production is not simply associated with reducing complexity. Referring to Figure 1, environmental and technological uncertainty is uncontrollable while network context is subject to exchange, and therefore in part controllable through negotiations. Therefore, in line with Engelseth et al. [5], it is suggested in healthcare service development to instead focus on developing features of the production context. This involves features of the firm’s resources, including its relationships. At an inter-organisational level, developing other firms’ resources and how these may be available to the healthcare process in question is also an important topic on the process development agenda. Change to enhance value creation should then focus on supporting actors in navigating through processes and their network context—maps for action.

## 6. Framing Healthcare Services

Compared to deterministic planning paradigms, an alternative way to analyse and practice healthcare service provision is to conceptualise healthcare as an emergent managerial process. This implies focus is directed to administering healthcare services and not the production itself. This invokes a view that it is the exchange economy we are taking into consideration. As previously discussed, this view of value creation does not dwell and act in isolation; it has a vital supporting function to producing healthcare processes. It is, however, our chosen research focus and approach. In healthcare, however, professionals both manage and produce. Therefore, this classification’s borderline when looking at individual actors, between those who manage and those who produce, is blurry.

This section, where we probe into the nature of framing healthcare services, is grounded in a balanced manner on literature from both economics and sociology. Through this, attention is pointed towards actors on how they think and behave in a networked economic context to achieve sustainable healthcare service provision, from institutional economics understanding regarding features of the healthcare service providing institution and reasoning to integrate with business relationships paired with sociological reasoning for how people perceive the economic activity that healthcare production is. This approach pairs this with the previously introduced ecosystems thinking. This approach is a direct adaptation to an approach developed from exporting by Engelseth and Glavee-Geo [4]. In this paper, attention was directed to framing managerial activity associated with exporting as an emergent process contingent on an unruly market context.

The sociologist Goffman [42] introduces “framing” to analyse how actors handle uncertainty in social activity. Callon ([43] p. 249) states that: “The frame establishes a boundary within which interactions–the significance and content of which are self-evident to protagonists–take place more or less independently of their surrounding context”. Using “framing” to study and analyse healthcare services involves directing focus towards the actors involved in managing and producing this service. This includes individuals, more or less integrated groups or teams, and the relationships bonding these entities. In this picture, frames concern the more or less continuous mental model of these actors, in this case, concerning healthcare provision.

Through the framing of healthcare services, our approach directs attention to the micro-level of such production. We argue that is part of the managerial aspect of producing since these mental models shape decisions. Decision-making as a sum of activities over time creates a health care service provision pattern. Thereby elevating the importance of human experience, the accumulated and therefore remembered cognition of managing healthcare. Learning is thus a key element here. When describing the healthcare services rendered by any involved institution, this framing represents a stable pattern perception. Eventually, to the degree that this framing is shared, it creates a culture of producing healthcare services. Although producing is dynamic, the mental models guiding production are more stable and, therefore, a new and highly potent dimension to the network context’s structure. It includes features of not only integration and interdependency but also features of network culture. Such culture may be nested in subcultures, as different networking actors (groups of people in an institution) may vary. Since we already have presumed that the route to healthcare process development goes through changing the network structure (context), this adds a new dimension to improving these services.

Then, how do we address healthcare service development through an analytical focus on how people frame activity. One key feature is to address how these professionals learn in the workplace. This learning includes the borders of activities in the relationship. Who does what when doing healthcare services? Applying the sociological concepts of roles adds conceptual crispiness to “framing”. Roles address the world of theatrical performance. They are associated with expectations of others to one who performs. This adds another dimension to interaction. This is not simple mechanical activity. It is a process where actors are somewhat tightly interwoven into a web of expectations. This picture is far from deterministic since this interaction may vary, and likewise, the cognition of the mental model, the framing, is never completely clear-cut. Through a relatively stable mental model, the framing by a performer develops through interaction and is enacted by the environment. Framing may, however, be disturbed; the picture may leak. Callon [43] terms this cognitive leaking as “overflowing”.

Framing involves the idea that a market is understood as “…a system of relationships between agents” ([43] p. 251)—the human actor, including its life experiences and perceptions of these, shape human action. “Framing defines the effectiveness of the market because, in this closed interactional space, each individual can take into account the viewpoint of every other individual when reaching a decision” ([43] p. 251). Some readers may react to describing the institutionalised context of healthcare service provision as “market”. This view is, however, in line with the basic assumptions of this paper. We view healthcare service as fundamentally an economic activity associated with resource economising to produce and capture value—aiming towards healthcare service use.

Just as the institutional economist Williamson [44,45] is concerned with how market imperfections blur rational economic behaviour, the sociologist Callon [43] points to how management is never a perfect decision-making process that follows the strict rules of rational behaviour. He cites two imperfections that prevail in interaction. Perfect framing may be challenging to achieve since business practice cloaked in varying uncertainty. Besides, Callon [43] discusses how framing also can be deliberately transgressed. This is what Williamson [44,45] classifies as opportunistic economic behaviour. This reasoning is associated with business ethics, and therefore organisational culture. Håkansson and Snehota [32], in the realm of network thinking, discuss how as integration develops, trust also grows, and thus such opportunism may cease. Williamson [44,45] does not discuss this idea of improving the problem of opportunism. This aspect of development is essential since economic misbehaviour is costly and leads to inefficiencies. It is a source of waste. Overflowing represents an array of process imperfections associated with the exchange economy.

## 7. Modelling How to Produce Sustainable Healthcare Services

The following provides a conceptual model for carrying out an in-practice inquiry about sustainable healthcare services applying process thinking. This form of process thinking directs contemplation towards the exchange economy. It directs attention to how, when leading and doing healthcare, people frame and act on these frames in an often-hectic everyday workplace setting. We regard sustainability as one of the critical factors to consider when analysing healthcare. This is because healthcare cannot be limited to a simple economic pattern of activity. It is an integral part of society and nature. Therefore, the studied process is involved in a very dynamic setting, the ecosystem. Following the concentric circles’ mode of sustainability (Figure 3), economic concerns are nested at the centre of healthcare production and encircled by layers of societal and natural environment concerns. This view depicts that any healthcare service provider’s immediate concern is to solve how to produce quality healthcare economically. Based on preceding research on framing exporting behaviour [3], the following circular model for analysing healthcare as managerial activity encompassing the aspect of healthcare as production is provided (see Figure 6):

The model illustrates our view of the importance of attaining customer value as the prime objective of the economic behaviour at the core of healthcare services. Framing is dual; it is done by the producer and by the recipient. In healthcare, this “customer” is two-fold, both the patient and next of kin. Quality is expected, and its degree of fulfilment is perceived. Herein lies the notion of overflowing. Concerning the degree of overflowing, this leads to potential overflowing in the producer’s framing, the healthcare service provider. What is vital to understand is that process development is not solely a technical matter. There is an important cognitive phase where people negotiate meaning regarding the ongoing service production. This exchange activity is then the foundation for technical change in the process, both its structure and dynamic aspect, i.e., how the process is designed as a workflow.

The basic assumption behind this approach’s usefulness is that the organisation using this procedure to understand and develop its healthcare processes possess a learning organisation culture. Gavin [46] points out three stages of organisational learning as (1) cognitive, (2) behavioural, and (3) performance improvement. Then, to investigate such a process, various aspects of actor framing of healthcare processes must be accounted for. By process, we here mean activity. This implies that it is ways of producing this service as operations, repetitive activities. It is not a simple series of actions but also an activity carried out on a more or less routine basis. The following depicts suggested topics brought up through administering a qualitative interview guide with various informants involved in the same process:

From the patient and next of kin side:Framing production: model the service as expected and experienced.Overflowing: depict some narratives of quality experiences (good/bad).Framing exchange: experienced quality of interactions.What are the learning experiences?

From the production side:Framing production: model the workflow “as-is”.Framing network contingencies: model the network structure of this workflow.Framing the ecosystem: describe aspects of nature and society relevant to the studied workflow.Overflowing: depict some narratives of quality experiences (good/bad).Framing exchange: experienced quality of interactions to produce and adapt to patient/next of kin needs.What are the learning experiences?Framing process improvement: perceptions on how the healthcare service can be improved?

The topics together provide the basis for empirical data that will help practitioners better understand patient needs and better adapt to these needs by understanding their perceptions regarding how they produce healthcare services. It is imminent that as many persons involved in the studied healthcare service process be involved in interviews. This means that an important aspect of this research will be to find patterns as well as try to understand the reasons for the discrepancy in perceptions.

## 8. Concluding Remarks

As pointed out by White [3], addressing sustainability increases research complexity since it widens the realm of which healthcare management must take into consideration upon decision-making. The economic challenges of healthcare are embedded in a wider realm of society and nature, and especially prone to ethical consideration. Healthcare is not only important, it is also an inherently emergent process when focus is directed to how such services are produced. Achieving a good economy of healthcare processes is both complicated and complex. This posits framing meaning as the starting point of investigation. This sets in focus various mind-sets of networked people in healthcare services. This network includes various stakeholders including the patients, their next of kin and the service producers. This is a process where technicalities of producing are not the focus, but what the people involved perceive regarding what is being done.

From a process view, sustainability looks into how people frame reality to manoeuvre in the healthcare service network. This network consists of professionals providing the service as well as patients and next of kin. The next step in this endeavour is to administer and carry out research in practice. Being sustainable we envision healthcare service development as iterative and continuous. The framework will also be subject to refinement through its use.

A case study research strategy could fit well here to study a process in detail in its real-life context. This research strategy is, however not directly linked with process development in healthcare practices. A case study will normally provide documentation that enables practitioners to later improve the healthcare service.

Alternatively, development can be integrated into the framework. This creates methodological challenges. If the aim is simply to improve processes not addressing issues of trustworthiness and credibility, academic demands may be overseen. However, action research has developed into an academically sound and societally powerful methodology. Koshy et al. [10] provide detailed guidance in how to conduct action research in a healthcare service setting. It provides an opportunity of more fast healthcare process improvement, as well as demanding integration practitioners and academic researchers to min an iterative manner change how e.g., healthcare is provided. This certainly evokes great potential. Action research is also applicable in continuous improvement schemes. Research avoids getting forgotten in the drawers of academia. It is sustainable since it is sensitive to the complexity of academia meet practice in an iterative manner following a continuous timeline of change and improvement in healthcare viewed as an ecosystem.

The developed framework suggests mainly use of qualitative research methodology since interview guides will need to be adapted to the different informants. Quantitative methodology, such as surveys and analytics, may play a supporting role in this research. It is also possible that simulation, e.g., applying agent-based modelling may be recommended at later stages of research, to test out new systems of healthcare before implementation. Using simulation related to our developed approach should fundamentally regard the applications as complex systems.

This approach points to both new ways to research as well as develop healthcare service practices. Importantly, it evokes a way to confront and depict the friction between patients/next of kin perception with the perceptions held by professionals producing the service. Intuitively this should lay grounds for negotiating to improve healthcare services based on how the service is expected and how practice is experienced. As stated in the introduction, the fundamental contribution is to view healthcare as a framed sustainable environmentally contingent managerial activity, fundamentally viewed as an emergent exchange process in an ecosystem context. Production solutions are designed and supported by this process laden with uncertainty in its practice. Managers’ aim in healthcare is not to demise this uncertainty but rather, learn how to live and cope with it. This is where our focus on framing and the mental model of healthcare practitioners and planners, represents in our view, a keystone to service development and management, is thus rendered as “navigation”.

## Figures and Tables

**Figure 1 ijerph-18-06336-f001:**
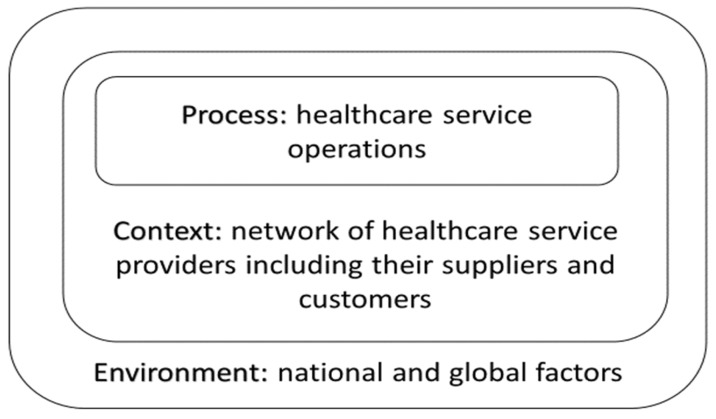
The healthcare service process as embedded unit of analysis [2].

**Figure 2 ijerph-18-06336-f002:**
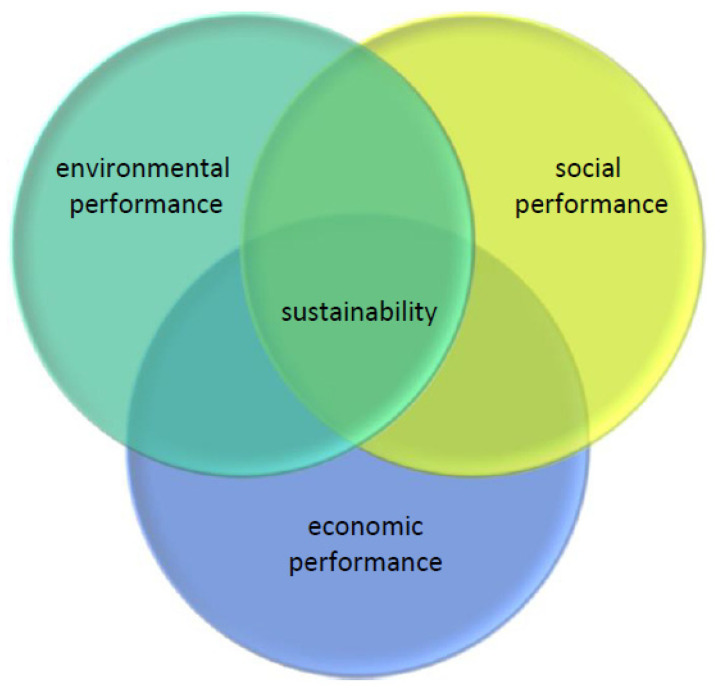
Interlocking circles model of sustainability [20].

**Figure 3 ijerph-18-06336-f003:**
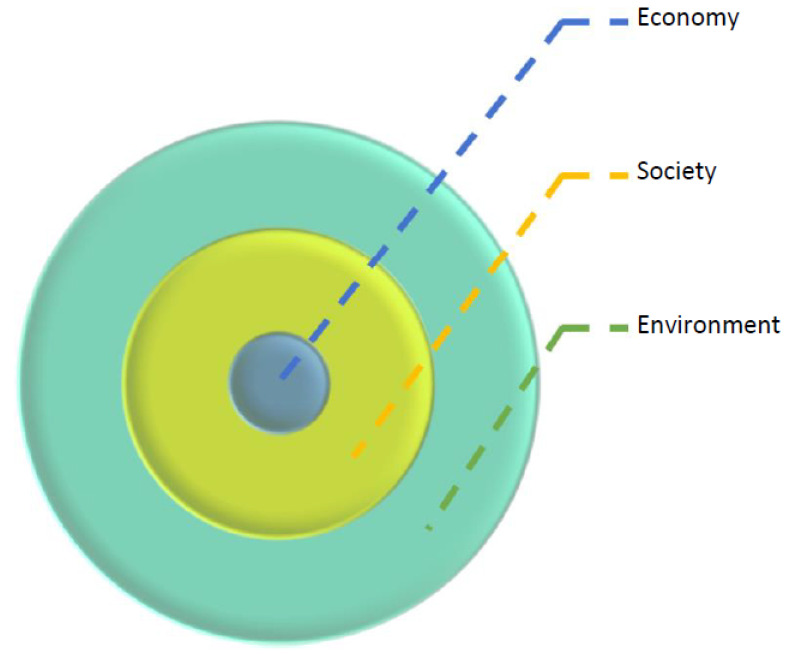
Concentric circles model of sustainability [20].

**Figure 4 ijerph-18-06336-f004:**
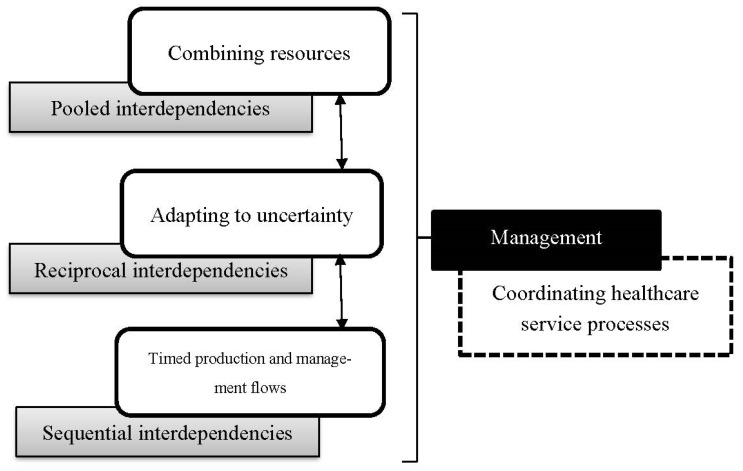
Interdependency and coordination of engineering processes. Adapted from Engelseth et al. [29].

**Figure 5 ijerph-18-06336-f005:**
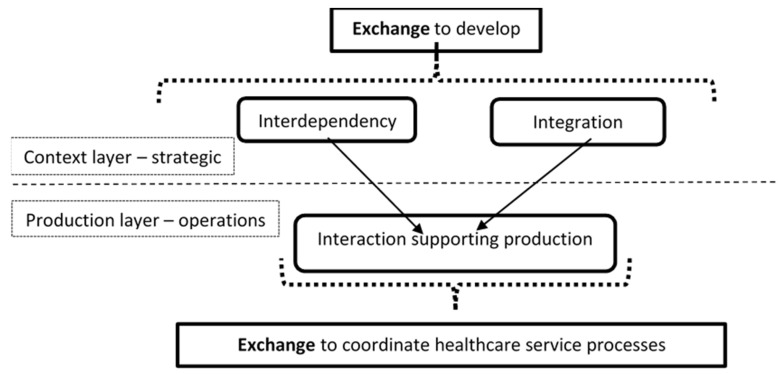
Interdependency, integration, and integration in a service network. Adapted from Engelseth et al. [27].

**Figure 6 ijerph-18-06336-f006:**
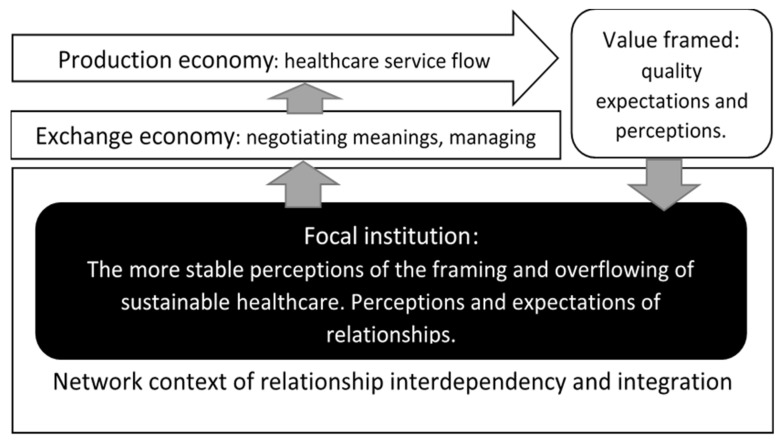
Conceptual model of dynamically managing healthcare service provision.

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
