# Peer review of "Framing Sustainable Healthcare Services"

_ijerph, 2021, doi:10.3390/ijerph18126336_

Round 1
Reviewer 1 Report
This is an interesting title, Framing Sustainable Healthcare Services and the abstract presented seems promising initially.
However, after reviewing the draft manuscript I am afraid that the quality and scope of the work done may not be sufficient for publication in this journal.
The paper had aimed to develop an analytical framework using process thinking to achieve sustainable healthcare services, however the work presented needs to be further improved.
The methodology employed in this conceptual framework appears fuzzy and have not been grounded on key past works related to sustainability in healthcare services.
The literature cited in parts have not been directly related to healthcare services. The conceptual framework must be supported by existing and established literature since it is conceptual.
No preliminary data or checks were presented as part of the development process for the model.
I will highly recommend the authors relook into their development process and attempt for future considerations.
Author Response
Thank you for your review. Although critical, we have taken note of your suggestions. We have revised the entire text to improve readability. The aim of this paper is to guide forthcoming qualitative research based on our epistemological stance. We create practical guidelines. The conclusion has been strengthened by a discussion of this approach in relation to implementation of developments related to both using action research and the more widespread case study research strategy. We have also critically checked for readability and amended phrasing.
Reviewer 2 Report
The topic is very interesting and, in my opinion, the findings are interesting. Please, if it is possible, remove self-citation to one or more contributors to this manuscript. For the future, I can suggest you extend the research with an interventionist approach in order to investigate the impact of your considerations on health care organizations.
Authors must work hardly to improve the paper by using a different research technique, for example the interventionist research (or Action research). In this light, the paper need of a rethinking, starting ny actual structure.
The Manuscript ID ijerph - 1218614 provides evidence of a concern in the process management of healthcare organization, highlighting one of the most relevant problem in these organizations, that is the analysis of the healthcare processes. I know that this manuscript is not a quantitative research paper, however I think that it could be a good starting point to improve the paper with an interventionist approach (see Action Research).
It seems a seminal research but well organized. If it could be useful, I can suggest Authors to improve their paper using such approach and providing empirical evidence of the impact of the arguments on healthcare efficiency.
Please, avoid self-citations.
Author Response
Thank you for the review. We have critically reviewed self-citations. We admit there are a few, but these are necessary in the text since this work builds upon our previous works. These citations are accordingly the conceptual foundation of this paper.
This is a conceptual paper aimed at guiding research practice, the methodology adapted to the specific healthcare management context applying our particular epistemological standing. However your suggestion to include a discussion is taken note of, and use of action research as well as case study research strategy related to healthcare process development is discussed now briefly in the conclusion. We clearly see this strengthens the potential impact of the paper to guide forthcoming research.
Reviewer 3 Report
You have very nicely laid out your conceptual framework, however, as always it is helpful if you provide a few examples (or case studies) of how it will work in the real world.
As always, add a section on implications to policy.
Meaning unclear in line 29, perhaps explain the meaning of being environmentally contingent
line 52 point out rather than point to?
I did find the paper hard going and when the authors come back with revisions I will probably make further suggestions for improvement. Probably a glossary of terminology should be added, etc.
Author Response
Thank you for the review. We have amended line 29 and 52. We have tried to make through this revision the wording applied in the text more self-explanatory as well as generally improve the writing style.
Reviewer 4 Report
See attached PDF

Author Response
Thank you for your review. We have in accordance with your major suggestion rewritten parts of the conclusion adressing the issue of practical impact of our approach developed thruogh the paper. This uincludes noiw also a brief discussion of how this approach relates to both case studies and implementation as well as a more integrated action research strategy.
Round 2
Reviewer 1 Report
Thank you for the resubmission. As in the original draft you are right in highlighting the complexity of the healthcare service system and the importance for the consideration of sustainability aspects.
However, the changes made through the addition of some points in the manuscript still have not been sufficient to address the limitations that i have notice in the first draft. I have not seen an effort of a major revision to the research process and draft that addresses my comments effectively. This is especially on the coverage of literature and that the framework is not grounded or designed to address specific problems in healthcare. I would recommend the team work closely with specific healthcare providers or experts in the development of the future framework. Good luck!
Author Response
Thank you for your advice. We have added a section both in the introduction as well as in the concluding remarks which adresses the problem of application in healthcare services. We also added a new referece from a recently published book which adresses the use of ecosystems in health care. We believe this should be sufficient.
Reviewer 2 Report
Dear Authors, the manuscript was improved clarifying some concern on the paper. In my opinion, I think you have discussed the framework clearly. I suggest avoiding self-citation if it is possible.
Author Response
Thank you for your valuable comments and approval.